

# Phylogeographic analysis and species distribution modelling of the wood frog *Batrachyla leptopus* (Batrachylidae) reveal interglacial diversification in south western Patagonia

José J. Nuñez[1], Elkin Y. Suárez-Villota[2], Camila A. Quercia[1], Angel P. Olivares[1] and Jack W. Sites Jr[3,4]

[1] Instituto de Ciencias Marinas y Limnológicas, Facultad de Ciencias, Universidad Austral de Chile, Valdivia, Región de Los Ríos, Chile
[2] Instituto de Ciencias Naturales, Facultad de Medicina Veterinaria y Agronomía, Universidad de Las Américas, Concepción, Región del Bio-Bío, Chile
[3] Department of Biology and M.L. Bean Life Science Museum, Brigham Young University, Provo, UT, United States of America
[4] Current affiliation: Department of Biology, Austin Peay St University, Clarksville, TN, United States of America

Corresponding author
José J. Nuñez, jjnunez@uach.cl

## ABSTRACT

**Background**. The evolutionary history of southern South American organisms has been strongly influenced by Pleistocene climate oscillations. Amphibians are good models to evaluate hypotheses about the influence of these climate cycles on population structure and diversification of the biota, because they are sensitive to environmental changes and have restricted dispersal capabilities. We test hypotheses regarding putative forest refugia and expansion events associated with past climatic changes in the wood frog *Batrachyla leptopus* distributed along ∼1,000 km of length including glaciated and non-glaciated areas in southwestern Patagonia.

**Methods**. Using three mitochondrial regions (*D-loop*, *cyt b*, and *coI*) and two nuclear loci (*pomc* and *crybA1*), we conducted multilocus phylogeographic analyses and species distribution modelling to gain insights of the evolutionary history of this species. Intraspecific genealogy was explored with maximum likelihood, Bayesian, and phylogenetic network approaches. Diversification time was assessed using molecular clock models in a Bayesian framework, and demographic scenarios were evaluated using approximate Bayesian computation (ABC) and extended Bayesian skyline plot (EBSP). Species distribution models (SDM) were reconstructed using climatic and geographic data.

**Results**. Population structure and genealogical analyses support the existence of four lineages distributed north to south, with moderate to high phylogenetic support (Bootstrap > 70%; BPP > 0.92). The diversification time of *B. leptopus*' populations began at ∼0.107 mya. The divergence between A and B lineages would have occurred by the late Pleistocene, approximately 0.068 mya, and divergence between C and D lineages was approximately 0.065 mya. The ABC simulations indicate that lineages coalesced at two different time periods, suggesting the presence of at least two glacial refugia and a postglacial colonization route that may have generated two southern lineages ($p = 0.93$,

type I error: <0.094, type II error: 0.134). EBSP, mismatch distribution and neutrality indexes suggest sudden population expansion at ∼0.02 mya for all lineages. SDM infers fragmented distributions of *B. leptopus* associated with Pleistocene glaciations. Although the present populations of *B. leptopus* are found in zones affected by the last glacial maximum (∼0.023 mya), our analyses recover an older history of interglacial diversification (0.107–0.019 mya). In addition, we hypothesize two glacial refugia and three interglacial colonization routes, one of which gave rise to two expanding lineages in the south.

# INTRODUCTION

The southern South American landscape is characterised by dynamic transformations resulting from tectonic processes and climatic cycles (*Ortíz-Jaureguizar & Cladera, 2006*, *Le Roux, 2012*). In particular, geological studies (*Mercer, 1972*; *Rabassa & Clapperton, 1990*; *Clark et al., 2009*) have demonstrated that the southwestern part of Patagonia has experienced at least four Pleistocene glaciations, including the most extensive Andean glaciation (1.1 mya), the coldest Pleistocene glaciation (0.7 mya), the last southern Patagonian glaciation (180 kya), and the Last Glacial Maximum (LGM; 20,500 and 14,000 years BP). It has been hypothesized that these climatic cycles re-organized ecosystem structures, altered species abundance and changed distribution patterns of many Patagonian taxa (*Sérsic et al., 2011*; *Giarla & Jansa, 2015*). It is also recognized that some areas served as climate refugia in a vast inhospitable region, and that those refugia provided habitat from which species expanded when environmental conditions were suitable (*Keppel et al., 2012*).

Phylogeographic studies in vertebrates and plants in this area (*Sérsic et al., 2011*) have highlighted the importance of such glacial refugia, where species survived through glacial maxima, and which today harbour high levels of genetic diversity and differentiated genetic clusters (*Ruzzante et al., 2006*; *Vidal-Russell, Souto & Premoli, 2011*; *Zemlak et al., 2011*). Postglacial colonization pathways have been also hypothesized for a range of species (*Victoriano et al., 2008*; *González-Ittig et al., 2010*; *Gallardo et al., 2013*; *Vidal et al., 2016*) to explain how extant populations are connected and how genetic diversity is spatially distributed.

Amphibians have attracted considerable attention on Pleistocene refugia hypotheses, largely due to their restricted dispersal capabilities that tend to facilitate allopatric differentiation (*Fitzpatrick et al., 2009*; *Carnaval et al., 2014*). Further, amphibians are highly sensitive to habitat disturbances owing to complex life histories, permeable skin, and exposed eggs (*Beebee, 1996*; *Prohl, Ron & Ryan, 2010*).

In Southwestern Patagonia, most of the amphibian species are endemic (70%) and strongly associated with humid Valdivian forest (*Formas, 1995*). These forests contracted

into smaller fragments during the more arid phases of the Pleistocene, leading to the isolation and allopatric diversification of forest-associated taxa (*Suárez-Villota et al., 2018*). One example is the grey wood frog *Batrachyla leptopus* Bell 1843. This small amphibian (30–35 mm snout vent length) lays eggs (diameter of ova 3–4 mm) in clusters of 93–146. Clutches are fertilized at the edges of small pools, amidst vegetation or under fallen logs and rocks on the ground, where embryonic development takes place (*Busse, 1971*; *Úbeda & Nuñez, 2006*). When autumnal rains flood the area (March-June), water stimulates hatching, and larvae metamorphose in 5–7 months (*Formas, 1976*). *Batrachyla leptopus* has one of the broadest distributions of any Chilean frog (*Cuevas & Cifuentes, 2010*), and is threatened by habitat deterioration in most of its geographic range. Furthermore, most of its current distributional area was intensively glaciated during the LGM but its genetic structure and the impact of the habitat lost are poorly known (*Heusser & Flint, 1977*; *Paskoff, 1977*).

Thus, while the humid ecological requirements of *B. leptopus* might in part explain its low abundance and patchy distributional pattern, Quaternary glaciations may have contributed to a phylogeographic history linked to glacial refugia. In fact, previous studies of *B. leptopus* (*Formas & Brieva, 2000*; *Vidal et al., 2016*) have revealed high levels of population divergence as a result of past climatic oscillations. For example, based on allozyme data *Formas & Brieva (2000)* inferred that the lack of correlation between genetic and geographical distances among *B. leptopus* populations could be the result of postglacial recolonization. On the other hand, *Vidal et al. (2016)* hypothesized that populations of *B. leptopus* originated from geographically differentiated gene pools, and specifically that post-glacial population expansions came from at least two refugia.

One important caveat for studies on refugial hypotheses is that they are often tested by revealing patterns of intraspecific relatedness of present-day biota and placing variation and divergence into a single evolutionary context. These approaches entail an inductivist point of view, that is, the view that researchers should first observe and analyse the present-day pattern and only then might explanations emerge in terms of historical processes (*Andersson, 1996*; *Arroyo-Santos, Olson & Vergara-Silva, 2014*; *Segovia & Armesto, 2015*). From an epistemological perspective, these approaches cannot progress beyond being speculative first attempts to understand the evolutionary history of a group, because they tend to generate, rather than test, hypotheses (*Crisp, Trewick & Cook, 2011*; *Papadopoulou & Knowles, 2016*). Approximate Bayesian computation (ABC) methods (*Beaumont, Zhang & Balding, 2002*) have introduced novel model comparison and parameter estimation in population genetic and phylogeographic studies (*Csilléry et al., 2010*; *Sunnåker et al., 2013*; *Inoue et al., 2014*). These methods provide an approximation of the posterior distribution of model probabilities and/or parameter values by simulating data with parameters drawn from specified prior distributions, and retaining values that produce data sets similar to the observed data to test alternative hypotheses (*Robinson et al., 2014*; *Freeland, 2020*).

Progress in phylogeographic studies has been further extended by the incorporation of species distribution modelling (SDM; *Phillips et al., 2017*). SDM approaches have been widely applied to assessment of species ranges, and to evaluate spatial and temporal hypotheses about current and past species occurrence (*Gavin et al., 2014*). Accessibility and

easy data requirements of correlative SDMs, coupled with the improved availability of paleoclimate simulations, have the special advantage of permitting prediction of distributional potential across scenarios of environmental change. These models are particularly relevant for understanding the effects that ongoing human-caused global climate change will have on biodiversity (*Wiens et al., 2009*), including the study of glacial refugia (*Gavin et al., 2014*).

In this work, we use a multilocus phylogeographic approach and species distribution modelling to test two independent hypotheses regarding putative forest refugia and expansion events in *B. leptopus*. We first test the hypothesis that *B. leptopus* postglacially colonized the southern area of its current distribution, and assess whether temporal and demographic patterns are consistent with such a scenario. If *B. leptopus* expanded postglacially from the LGM, then populations should exhibit the genetic signature of recent rapid expansion, and the divergence time between populations present in non-glaciated and glaciated areas should be consistent with such postglacial expansion. Second, we test the hypothesis that *B. leptopus* colonized southwestern Patagonia from a single refuge; if *B. leptopus* expanded southward from a single refuge, then genetic variation in southernmost populations is expected to be low. Alternatively, if *B. leptopus* expanded from multiple refugia, then greater genetic diversity would be expected in some populations representing suture zones where genetic admixture may have occurred, and several coalescent points should be detected.

To this aim, we first estimated the genetic structure among *B. leptopus* populations and reconstructed its phylogeographic relationships under maximum likelihood and Bayesian inference. Second, we estimated divergence times and temporal changes in population sizes to determine if these were consistent with late Pleistocene events. Then, we examined the demographic history of this species by simulating alternative Pleistocene glaciation scenarios in an ABC framework. Finally, we combined demographic inferences with species distribution modelling in *B. leptopus.*

## MATERIALS & METHODS

### Sample collection

Between 2009–2018 we collected 130 individuals and buccal swabs (most samples) from 19 localities throughout the distributional range of *B. leptopus* in south western Patagonia (Table 1; Fig. 1). Each sampling site was geo-referenced with a GPS Garmin GPSmap 76CSx. Eight individuals of *B. taeniata* were used as the outgroup. This study was carried out under supervision and approval of the Bioethics and Biosecurity Committee of the Universidad Austral de Chile (UACh, Resolutions No. 236/2015 and 61/15), and the Servicio Agrícola y Ganadero (SAG, Resolution No. 9244/2015).

### DNA extraction, amplification, and sequence alignment

Whole genomic DNA was extracted either from liver tissues or buccal swabs according to *Broquet et al. (2007)*, using the manufacturer's recommended protocol for the Qiagen DNeasy tissue kit (Cat. No. 69506). We amplified three mitochondrial regions: a segment of the Control region (*D-loop; Goebel, Donnelly & Atz, 1999*) Cytochrome *b* (*cyt b; Degnan &*

**Table 1 Sampling locations of *B. leptopus*, coordinates and sample size for each location (N), and lineage structure according to GENELAND.**

|  | Location | Latitude | Longitude | N | Lineage structure |
|---|---|---|---|---|---|
| 1 | Los Queules (LQ) | −35.99277778 | −72.52583333 | 12 | A |
| 2 | Nahuelbuta (NA) | −37.78861111 | −72.99222222 | 7 | B |
| 3 | Mafil (MA) | −39.66964722 | −72.92111111 | 2 | C |
| 4 | Pichirropulli (PI) | −40.13305556 | −72.90888889 | 7 | |
| 5 | Cordillera Pelada (CP) | −40.14027778 | −73.41777778 | 10 | |
| 6 | Bahía Mansa (BM) | −40.56305556 | −73.73166667 | 12 | |
| 7 | Antillanca (AN) | −40.66888889 | −72.16027778 | 3 | D |
| 8 | Alerce Andino (AA) | −41.58027778 | −72.54083333 | 5 | |
| 9 | Sarao (SA) | −41.16416667 | −73.72722222 | 3 | |
| 10 | Puntra (PT) | −42.10388889 | −73.87000000 | 11 | |
| 11 | Pumalín (PU) | −42.61638889 | −72.47805556 | 3 | |
| 12 | Tepuhueico (TP) | −42.71833333 | −73.94138889 | 11 | |
| 13 | Isla Lemuy (LY) | −42.62416667 | −73.63722222 | 6 | |
| 14 | El Amarillo (EA) | −42.89444444 | −72.46833333 | 14 | |
| 15 | Yaldad (YA) | −43.10527778 | −73.69555556 | 14 | |
| 16 | Marín Balmaceda (MB) | −43.78388889 | −72.96388889 | 1 | |
| 17 | La Junta (LJ) | −43.93944444 | −72.36361111 | 7 | |
| 18 | Lago Verde (LV) | −44.23472222 | −71.84083333 | 1 | |
| 19 | Queulat (QU) | −44.37694444 | −72.54138889 | 1 | |

*Moritz, 1992*), and Cytochrome oxidase subunit I (*coI*; *Folmer et al., 1994*), and two nuclear regions: Propiomelanocortin (*pomc*; *Gamble et al., 2008*) and *β* Crystallin A1 (*crybA1*; *Dolman & Phillips, 2004*), via polymerase chain reaction (PCR). Reaction cocktails for PCR were according to *Suárez-Villota et al. (2018)*. PCR products were sequenced in Macrogen Inc. (Seoul, Korea) and at the DNA Sequencing Center at Brigham Young University (Provo, USA). To transform sequence data to haplotypes we used PHASE v2.1.1 (*Stephens & Donnelly, 2003*) with the default model for recombination rate variation (*Li & Stephens, 2003*). We aligned sequences using the automatic assembly function in Sequencher v. 4.8 (Gene Codes Corp.), and inspected the aligned sequences by eye, and made corrections manually.

## Molecular diversity and lineage structure

Haplotype and nucleotide diversity indexes (*Nei, 1987*) and their standard deviations, were estimated with DNASP v5.0 (*Librado & Rozas, 2009*) using all markers. The possibility of saturation in the rate of base substitutions was assessed by the method of *Xia et al. (2003)* using DAMBE v6.0 (*Xia & Xie, 2001*). Population genetic structure was examined using the package GENELAND v4 implemented in R v3.1.2 (*Guillot, Mortier & Estoup, 2005*), to infer the number of populations by giving a spatial model of cluster membership without prior designations. GENELAND was run with a model of uncorrelated allele frequencies for the mitochondrial locus (with all gene regions concatenated). We performed eight independent runs of $1.5 \times 10^7$ iterations, with thinning set to 500 and a "burn in" of

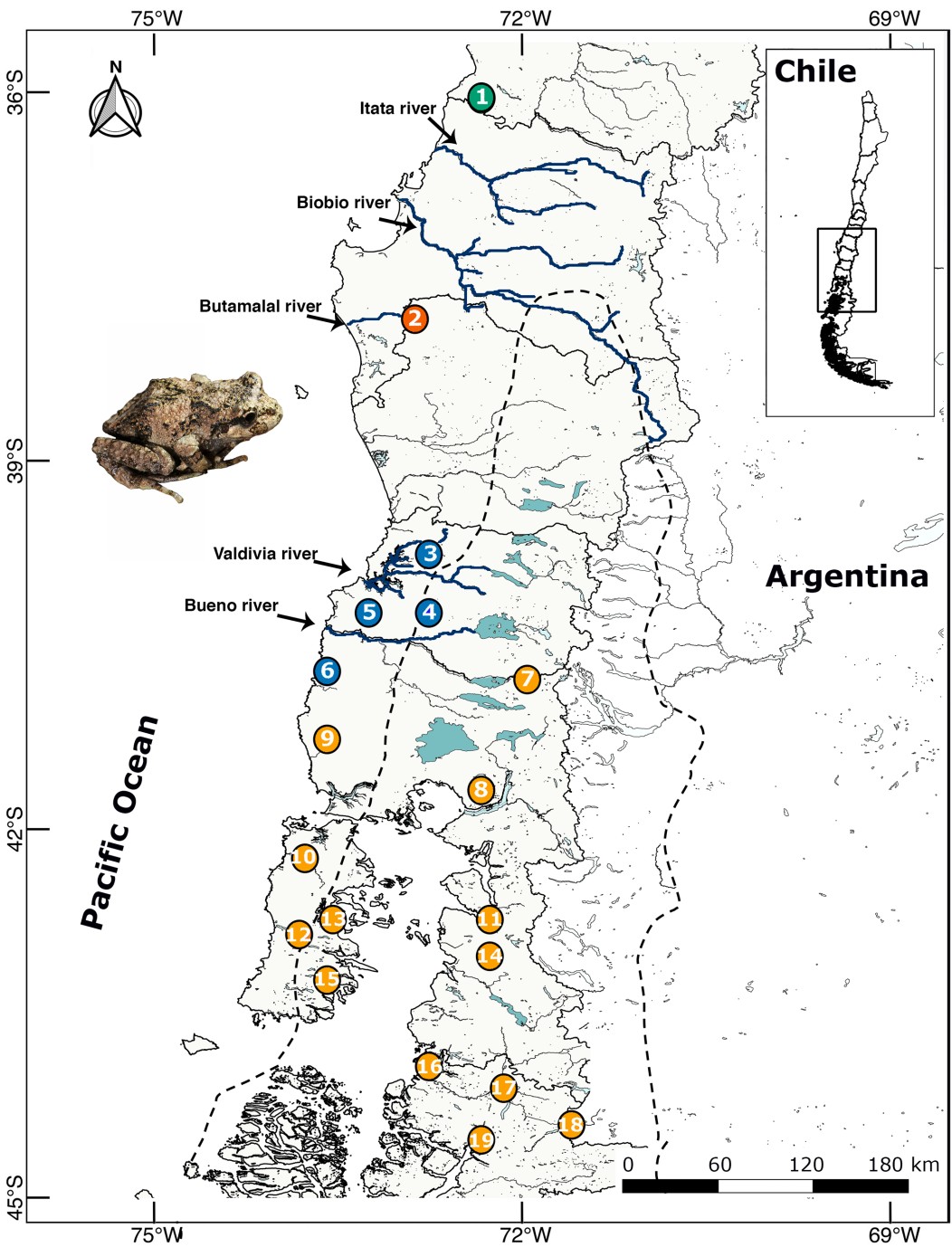

**Figure 1  Location map of 19 localities of *B. leptopus* sampled throughout the species' range in Southwestern Patagonia.** The dashed line corresponds to the limits of the LGM. Colours indicate different clades according the Geneland analyses. The full names of the populations are provided in Table 1.

20%. The number of possible clusters tested ranged from 1 to 19 (according to sampling locations). The level of population structure among the clusters obtained by GENELAND, was assessed by analysis of molecular variance (AMOVA; *Holsinger & Weir, 2009*) using ARLEQUIN v3.1 (*Excoffier, Laval & Schneider, 2005*) for mtDNA and nDNA separately. Also, using all loci we evaluated whether the sequences evolved under strict neutrality using Tajima's D (*Tajima, 1989*), Fu & Li's D (*Fu & Li, 1993*), and $r^2$ (*Ramos-Onsins & Rozas, 2002*) tests.

## Phylogenetic trees reconstruction, split networks and divergence time estimates

Previous to phylogenetic analyses, evolutionary models and partitioning strategies were evaluated using Bayesian information criterion (BIC) scores (*Schwarz, 1978*) in PARTITIONFINDER v2.1.1 (*Lanfear et al., 2017*) (Table S1). The phylogenetic analyses were performed with the combined mitochondrial and nuclear matrix. Partitioned maximum likelihood analyses were conducted using GARLI 2.0 (*Zwickl, 2006*) with 200 replicates of nonparametric bootstrap for branch support. Bayesian analyses were performed using MRBAYES v3.2 (*Ronquist et al., 2012*) with four independent MCMC runs of 50 million generations, sampling every 2,000 generations. Posterior distributions for parameter estimates and likelihood scores to approximate convergence were visualized with the TRACER program v1.6.0 (*Rambaut et al., 2014*). The effective sample sizes (ESS) of each parameter (>200) allowed us to confirm that samples were adequate for all analyises. A maximum clade credibility tree was visualized with the program FIGTREE v1.4.4 (http://tree.bio.ed.ac.uk/software/figtree/).

Posterior probability values >0.95 were taken as high statistical support for a clade being present on the tree (*Huelsenbeck & Rannala, 2004*). In order to obtain additional statistical support for the best tree obtained, topologies of different trees (ML and Bayesian) were compared with the use of the Shimodaira-Hasegawa (S-H) test (*Shimodaira & Hasegawa, 1999*) with resampling-estimated log likelihood (RELL), and bootstrapping of 1,000 replicates, using the program PAUP*.

We are aware that phylogenetic methods may not apply at the within-species level, due to multifurcating population genealogies in which descendant alleles coexist with ancestral ones, and recombination events may produce reticulate relationships (*Posada & Crandall, 2001*). To consider these caveats, we constructed unrooted phylogenetic networks using the method described by *Huson & Bryant (2006)*, implemented in SPLITSTREE v4.14.4.

To determine when major clades and lineages diverged relative to Quaternary glaciation history, we estimated time since the most recent common ancestor (TMRCA), using the reconstructed species tree from concatenated mitochondrial and nuclear sequences. For this reconstruction, we used the multi-species coalescent module implemented in *BEAST of BEAST v1.8.4 (*Drummond & Rambaut, 2007*; *Heled & Drummond, 2010*), and the same models used for phylogenetic tree reconstruction found by PARTITIONFINDER. Because that it is not possible to date any of the nodes within *B. leptopus*, as there are no fossils or dated biogeographic events, we used prior Neobatrachian mutation rates of 0.291037% per million years for COI, 0.37917% per million years for each other mitochondrial markers

(*dloop* and *cytb*), and a rate of 0.3741% per million years for *pomc* and crybA1 sequences, according to *Irisarri et al. (2012)*. Bayes factor analysis (*Li & Drummond, 2012*) indicated that species trees reconstructed under a strict-clock model received decisive nodal support compared to uncorrelated exponential or uncorrelated lognormal relaxed-clock models. Markov chains in BEAST were initialized using the tree obtained by MRBAYES, to calculate posterior parameter distributions, including the tree topology and divergence times. We used BEAST to estimate divergence times with runs for $2 \times 10^7$ generations, and sampling every 1000th generation. The first 10% of samples were discarded as "burn in", and we estimated convergence to the stationary distribution and acceptable mixing (ESS >200) using TRACER v1.6.0.

## Population-size dynamics through time

Hypotheses of historical demographic expansions and dynamics through geological time of the inferred lineages were estimated by "mismatch distributions" (*Rogers & Harpending, 1992*), and Extended Bayesian Skyline Plots (EBSP; *Heled & Drummond, 2008*) respectively. We used in a complementary way both approaches because small sample sizes apparently fail to provide enough power to Bayesian skyline plots to detect population expansion (*Grant, 2015*). The smooth, unimodal distributions typical of expanding populations can be readily distinguished from the ragged, multimodal distribution "signatures" of long-term stationary populations, by means of the 'raggedness' of these distributions (*Rogers & Harpending, 1992*). Confidence intervals for these estimates were obtained by simulations using the coalescence algorithm as implemented in DNASP v5.0. To estimate population-size dynamics through geological time, we reconstructed Extended Bayesian Skyline Plots (EBSP; *Heled & Drummond, 2008*) implemented in BEAST for the four lineages obtained with GENELAND. This coalescent-based, nonparametric Bayesian MCMC algorithm incorporates multi-locus data to reduce error estimates associated with single genes (e.g., traditional Bayesian Skyline Plots), and increases the power to resolve alternative demographic histories (*Ho & Shapiro, 2011*). For each EBSP, the appropriate model of nucleotide substitution was determined using PARTITIONFINDER. Genealogies and model parameters for each lineage were sampled every 1000th iteration for $2 \times 10^7$ generations under a strict molecular clock with uniformly distributed priors, and a "burn in" of 2000. Demographic patterns for each analysis were plotted in EXCEL v14.7.7.

## Test of phylogeographic hypotheses with ABC

A coalescent method was used to test phylogeographic hypotheses by constraining the genealogies to fit alternative evolutionary models, and assessing each model's fit by comparing the observed genetic pattern with the range of simulated patterns. Competing phylogeographic hypotheses were compared using an approximate Bayesian computation method (ABC approach), as implemented in DIYABC v2.1 (*Cornuet et al., 2014*). We evaluated five demographic scenarios to test alternative divergence times and tree topologies of the four main lineages recovered in GENELAND and in the phylogenetic analyses. Furthermore, the divergence of the main four lineages occurred before the last glacial maximum so the divergence scenarios were proved in such range. The refugia

hypotheses correspond to the points of coalescence, with four lineages strongly supported, the possibilities of coalescence are from 1 to 3, for which we tested the possibilities except scenarios with postglacial admixture, because there are no- paraphyly events between lineages. The prior coalescence points at time times t1 and t2 applied in the ABC correspond to those estimated by BEAST for the origin of *Batrachyla leptopus* (t2), and the divergence of the four lineages (t1, lower and higher range of the four lineages). Thus, all historically relevant scenarios differed only in the order of population divergence, and in the number and timing of demographic expansion events. These alternatives were: Scenario 1—the null model—all four lineages coalesced at t1 with equal divergence rates. Scenario 2—also a null model, but all four lineages coalesced at t2 with equal divergence rates. Scenario 3—the first coalescence of lineages A and B at t1, whose ancestor coalesced at t2 with lineage C and D. Scenario 4—the first coalescence of lineages C and D at t1, whose ancestor coalesced at t2 with lineage A and B. Scenario 5—one split event at t1 isolated the north (lineages A and B) from the south (lineages C and D) clades, and then a coalescence of both clades at t2 (see Results). We tested other scenarios whose divergence time were older and the probability was very low so they were not considered.

Prior values of *Ne* were set as 1,000–500,000 individuals with a uniform distribution, based on *Ne* calculated from MIGRATE-N v3.6 (*Beerli, 2006*). We performed maximum likelihood using 10 short chains of 1,000 steps, and two long chains of 10 000 steps, sampling each 100 steps, and a burn-in of 10 per cent. Ne was calculated using mitochondrial and nuclear rates reported by *Irisarri et al. (2012)*.

Prior values for divergence of the ancestral populations were based on divergence times calculated here (see Results), and a generation time of 2–3 yr (*Martin & Palumbi, 1993*), using a uniform distribution. Divergence times were set at between 20,000–500,000 generations ago for t2, and 10,000–200,000 generations for t1.

## Paleo-distribution and species distribution modelling

A total of 120 occurrence records were used for the species distribution models (SDMs) and paleo-distribution modelling. Records were obtained from peer-reviewed literature, our sampled sites, and online databases (GBIF: gbif.org, VetNet: vertnet.org, and iDigBio: idigbio.org). We modeled the SDMs using the standard 19 bioclimatic variables downloaded from Worldclim (*Hijmans et al., 2005*) for the current conditions (1960–1990), the Mid Holocene (Mid-Hol, ∼6,000 yrs BP), the LGM (∼22,000 yrs BP), and the Last Inter-glacial (LIG, ∼120,000–140,000 yrs BP). The variables were at 30 arc-sec for the current conditions and the LIG, and at 2.5 arc-min for the Mid-Hol and LGM. The Mid-Hol and the LGM variables were based on the Community Climate System Model (CCSM) and the Model for Interdisciplinary Research on Climate (MIROC), and while LIG conditions were based on *Otto-Bliesner et al. (2016)*. We restricted the projection of the models by creating a buffer of 2° around the outermost occurrence records and the known distribution of *B. leptopus*. All SDMs were performed using MAXENT v3.4.0 (*Phillips et al., 2017*). To avoid model overfitting and account for the correlation between the variables and the presence of outliers encountered during data exploration, we reduced the number of variables to five. This was performed by retaining the variables with |rho| < 0.8 that

**Table 2 Genetic diversity by lineage on *B. leptopus*.** N, Sample Size; H, Haplotype number; S, Segregating sites; Hd, Haplotype diversity; Pi, Nucleotide diversity; Neutrality test indexes (Rozas' $r^2$, Tajima's D and Fu's FS).

| Lineage | N | H | S | Hd | Pi | $r^2$ | D | Fu's FS |
|---------|-----|-----|-----|-------|---------|------------|-----------|------------|
| A | 12 | 12 | 58 | 1 | 0.00485 | 0.1501[**] | −1.50321[*] | −1.81336[*] |
| B | 7 | 2 | 2 | 0.476 | 0.00035 | 0.2381[**] | 0.68731[*] | 1.14506[*] |
| C | 31 | 29 | 127 | 0.996 | 0.00891 | 0.0823[*] | −1.05159[*] | −1.77929[*] |
| D | 80 | 72 | 150 | 0.998 | 0.00999 | 0.0651[*] | −1.26706[*] | −1.34060[*] |
| Total | 130 | 113 | 264 | 0.996 | 0.01448 | 0.0541[*] | −1.48283[*] | −2.26131[*] |

**Notes.**
[**] $P < 0.02$.
[*] non significant, $P < 0.10$.

contributed the most to ten cross-validated models, as shown by a Jackknife test. These variables were Bio2 = Mean Diurnal Range, Bio4 = Temperature Seasonality, Bio5 = Max Temperature of Warmest Month, Bio13 = Precipitation of Wettest Month, and Bio17 = Precipitation of Driest Quarter.

For all models, the equal training and sensitivity threshold rule was applied and the cloglog output was selected. Extrapolation was not used and clamping was applied when hind-casting the model of the current conditions to the past. The models for the Mid-Hol and for the LGM were overlaid within each time period to identify areas of agreement and disagreement between the models for each the time period. All models were transformed to binary using the selected threshold rule (*Phillips et al., 2017*).

# RESULTS

## Genetic structure using Bayesian clustering model

A total of 113 haplotypes were found when mitochondrial and nuclear markers were combined (Table 2). Saturation tests as a function of the genetic distance estimated under substitution model GTR showed non- significative saturation of DNA sequence alignments. The mtDNA Bayesian analysis with GENELAND yielded a modal number of four clusters ($K = 4$), recovered from all independents runs (Fig. 2A); this is based on the highest average posterior probability. The distribution of these four clusters, from north to south, was named as follows (Fig. 2B): lineage A (Los Queules), lineage B (Nahuelbuta), lineage C (Bahía Mansa, Cordillera Pelada, Máfil, and Pichirropulli), and lineage D (all remaining localities shown in Table 1). The highest estimate of haplotype diversity was found in lineage A, whereas the lowest values were found in lineage B (Table 2). Highest nucleotide diversity was detected in lineage D and the lowest one in lineage B (Table 2). Negative but non-significant values for Tajima's D and Fu's FS neutrality test indexes were found in all lineages except for B lineage. Here the non-significant positive values likely reflected the smallest sample size for this locality. Rozas's $r^2$ were positive for all lineages but only significant for A and B lineages, suggesting recent expansion (Table 2).

The AMOVA results using the four lineages indicate a significant genetic structure (groups defined as Lineages A, B, C, and D): (1) variation among lineages = 35.64% and variation within lineages = 45.93% for mtDNA; (2) variation among lineages = 0% and

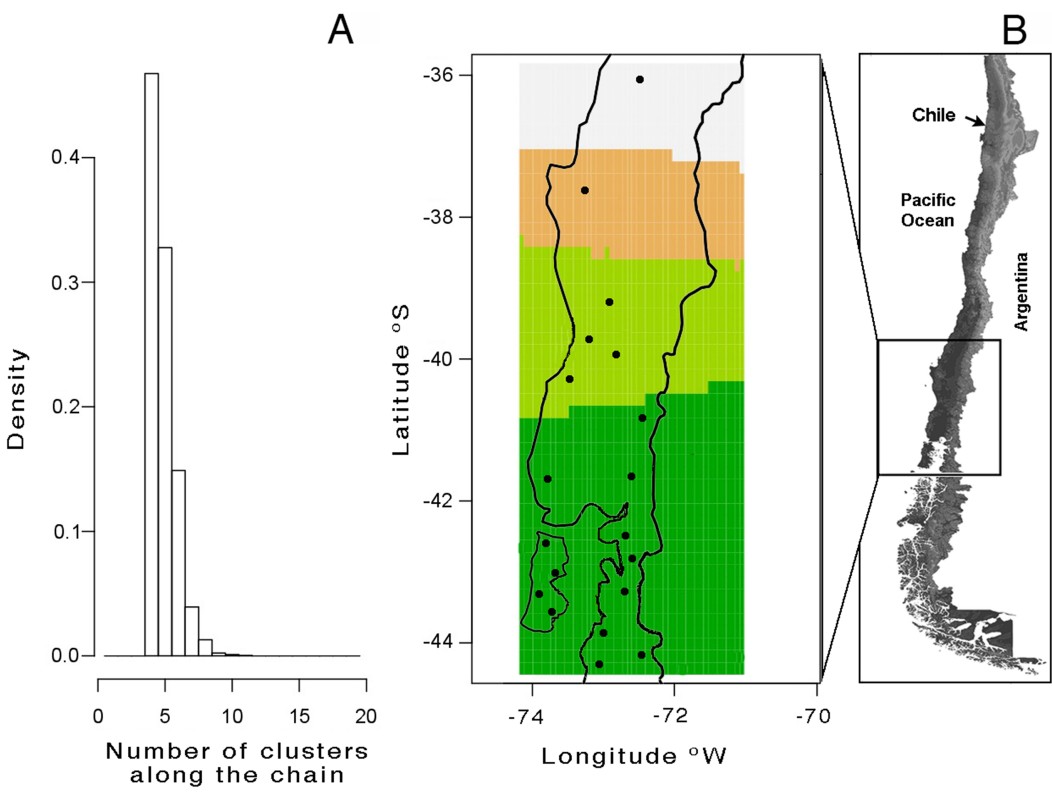

**Figure 2  Genetic structure analyses of *Batrachyla leptopus* populations.** (A) Number of populations simulated from the posterior distribution obtained with GENELAND v4.0; (B) spatial distribution of clusters. Black points correspond to sample sites. Lineage A to D are represented by light grey, orange, light green and dark green, respectively.

variation within lineages = 51.12% for *pomc*; (3) variation among lineages = 4.49% and variation within lineages = 2.74% for *crybA1* (Table 3).

## Phylogenetic tree reconstruction, split networks, and lineage divergence time

The models selected for the ML and Bayesian analyses are described in Table S1. Because the Bayesian analyses recovered a maximum clade credibility tree similar to the best ML tree, and the Shimodaira-Hasegawa test showed that topological disagreements were restricted to "low-support" nodes, we show only the Bayesian tree (Fig. 3A). The same four lineages recovered by GENELAND were recovered in the phylogenetic reconstruction with moderate to high support values. Lineages A and B were recovered as sister groups (bootstrap = 96%; BPP = 0.99), as were the C and D lineages, but with moderate support (bootstrap 74%; BPP= 0.95). Mitochondrial phylogenetic analyses recovered similar results to concatenated datasets, but the nuclear phylogeny was highly polytomized (Fig. S1). The split networks (Fig. 3B) recovered the same four lineages obtained by the ML and Bayesian analyses. The fit index was 94.96, meaning that only 5.04% of the distances in the distance

**Table 3 Results of hierarchical analysis of molecular variance for *Batrachyla leptopus* lineages, over nuclear and mitochondrial markers.** Results of hierarchical analysis of molecular variance for *Batrachyla leptopus* lineages, over mitochondrial, *pomc*, and *crybA1* markers, respectively. df, degrees of freedom; SS, sum of squares; *p*-value is based on 1,000 permutations.

| Source of variation | df | SS | Variance components | Percentage of variation | *p* |
|---|---|---|---|---|---|
| *Mitochondrial* | | | | | |
| Among lineages | 3 | 1,781.206 | 16.16587 | 35.64715 | 0.034 |
| Among localities within lineages | 15 | 2,036.172 | 20.83160 | 45.93547 | <0.001 |
| Within localities | 111 | 926.981 | 8.35222 | 18.41738 | <0.001 |
| *pomc* | | | | | |
| Among lineages | 3 | 32.023 | 0 | 0.00000 | <0.001 |
| Among localities within lineages | 15 | 148.948 | 1.38080 | 51.12677 | <0.001 |
| Within localities | 111 | 164.776 | 1.48447 | 54.96532 | 0.293 |
| *crybA1* | | | | | |
| Among lineages | 3 | 0.738 | 0.00488 | 4.49187 | 0.212 |
| Among localities within lineages | 15 | 1.785 | 0.00298 | 2.74255 | 0.153 |
| Within localities | 111 | 11.185 | 0.10076 | 92.76558 | 0.193 |

matrix are not represented by the network. Most of the internal splits have bootstrap support between 68 and 100%.

Divergence dating indicates that the A–B and C–D clades separated during the late Pleistocene, approximately 0.107 mya [95% confidence interval (CI) = 0.020–0.278 mya]. The divergence between A and B lineages would have occurred by the late Pleistocene (approximately 0.068 mya; 95% CI [0.036–0.147] mya) and divergence between C and D lineages was approximately 0.065 mya (95% CI [0.056–0.092] mya) (Fig. 3A).

## Demographic patterns of the inferred clusters

Results of mismatch distribution analyses (Figs. 4A–4D) revealed a single primary peak for lineage A, but with a non-significant raggedness index ($r = 0.0230$, $P > 0.1$). Similarly, unimodal patterns were observed in lineages C ($r = 0.003$, $P < 0.001$) and D ($r = 0.0008$, $P < 0.001$). The small sample size ($n = 7$) and haplotype numbers ($H = 2$) precluded this analysis on lineage B. Reconstruction of the demographic histories by means of Extended Bayesian Skyline Plot (Figs. 4E–4H) suggested population expansions for all lineages except B (Fig. 4F). EBSP further resolved sequential demographic expansions from the oldest, lineage D (c. 18,000 years bp), then lineage A (c. 11,000 years bp), and most recently, lineage C (c. 5,000 years bp).

## Hypothesis testing with ABC

Logistic regression analysis with DIYABC identified Scenario 4 as most strongly supported among the five tested (Fig. 5D), with a high posterior probability (0.93; Table 4); all other scenarios had much lower support (0.0–0.04). Moreover, the Type I and Type II error rates estimated for Scenario 4 were the lowest in both cases (0.14; 0.014–0.196; Table 4).

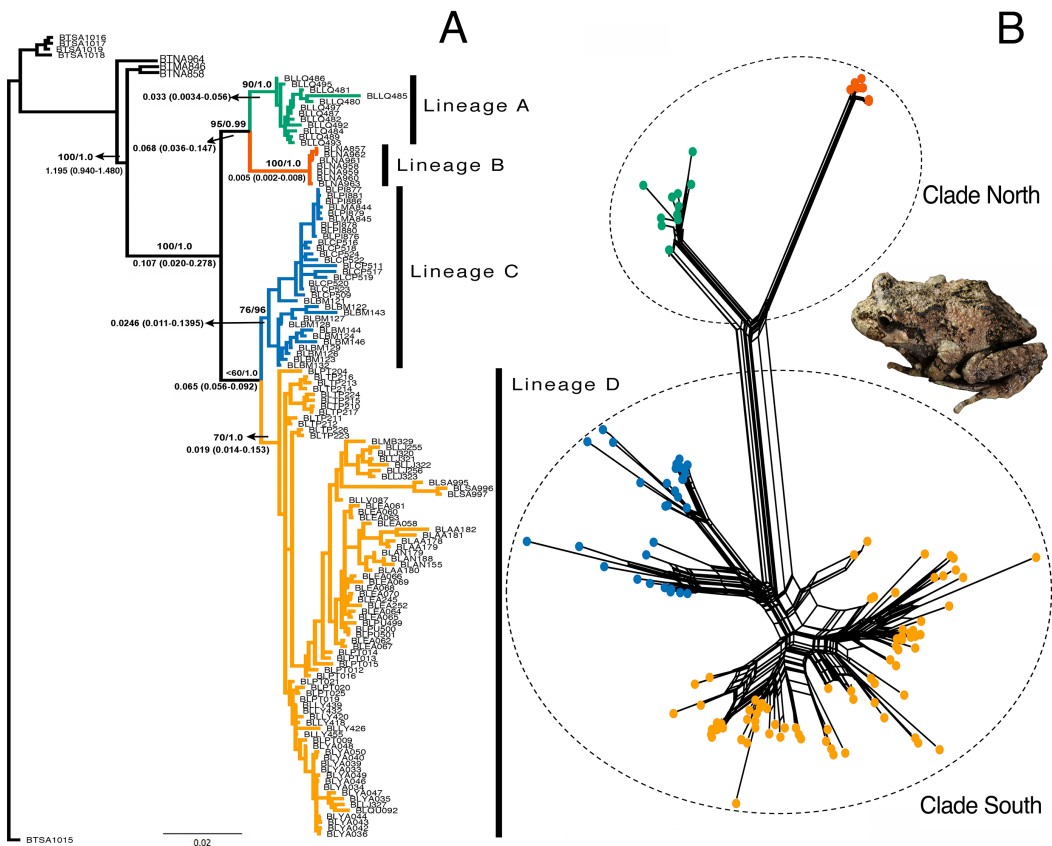

**Figure 3** **Phylogenetic reconstruction of *Batrachyla leptopus*.** (A) Bayesian tree of pooled data. Strongly supported lineages are represented with different colours and as lineages A–D. Branch support is based on ML bootstrap resampling and Bayesian posterior probabilities above the branch, and point estimates of selected divergence estimates are below. (B) Genealogical relationship based on unrooted phylogenetic networks showing the north and south clades.

Scenario 4 placed the first divergence as the split between lineages A, B, and the ancestor of the southern clade (lineages C and D) at t2, and the second split between lineages C and D at t1 (Fig. 5D). The effective population size (*Ne*) and divergence time parameters, in terms of the number of generations (t), estimated for this divergence scenario (Table S2), corroborate the population expansions inferred by EBSP and mismatch distribution for lineages C and D.

## SDMs and paleo-distribution models

The predicted distribution models of *B. leptopus* under four periods (last inter-glacial to current) are shown in Fig. 6. The model for the current conditions showed that the distribution of this species is mostly encompassed by the county-based current distribution mapped by the IUCN, with a high AUC value (0.961, SD = 0.013) (Fig. 6A). The two circulation models for the Mid-Hol showed different distributions: the MIROC-based model showed a continuous distribution similar to the current conditions model, while the CCSM-based model resolved disjunct distributions for the northern region of the current

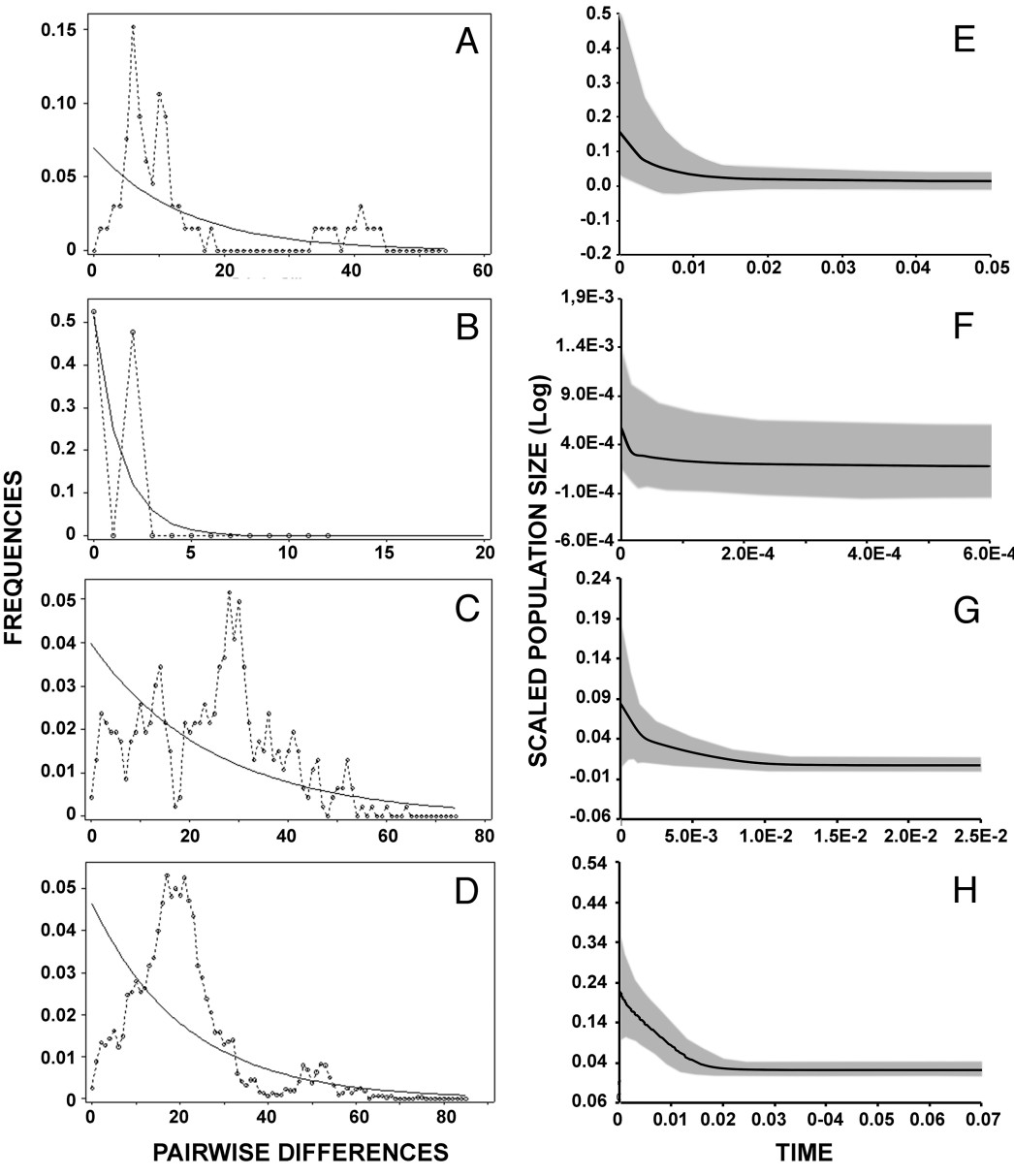

**Figure 4** **Historical demographic analysis for each lineage of *Batrachyla leptopus*.** (A–D) Mismatch distribution of observed frequencies of pairwise differences among *B. leptopus* lineages for concatenated data. (E–H) Extended Bayesian skyline plots analysis. (A, E) Lineage A. (B, F) Lineage B. (C, G) Lineage C. (D, H) Lineage D. X-axe corresponds to time (Ma) and y-axe corresponds to *Ne*, the product of effective population size and generation length in years.

distribution model (Fig. 6B). Both LGM models showed clear disjunct distributions with concordance around Nahuelbuta mountain range (Fig. 6C). The MIROC based model indicated a distribution to the north of the current distribution model, and the CCSM resolved a highly fragmented distribution. The model for the LIG showed a

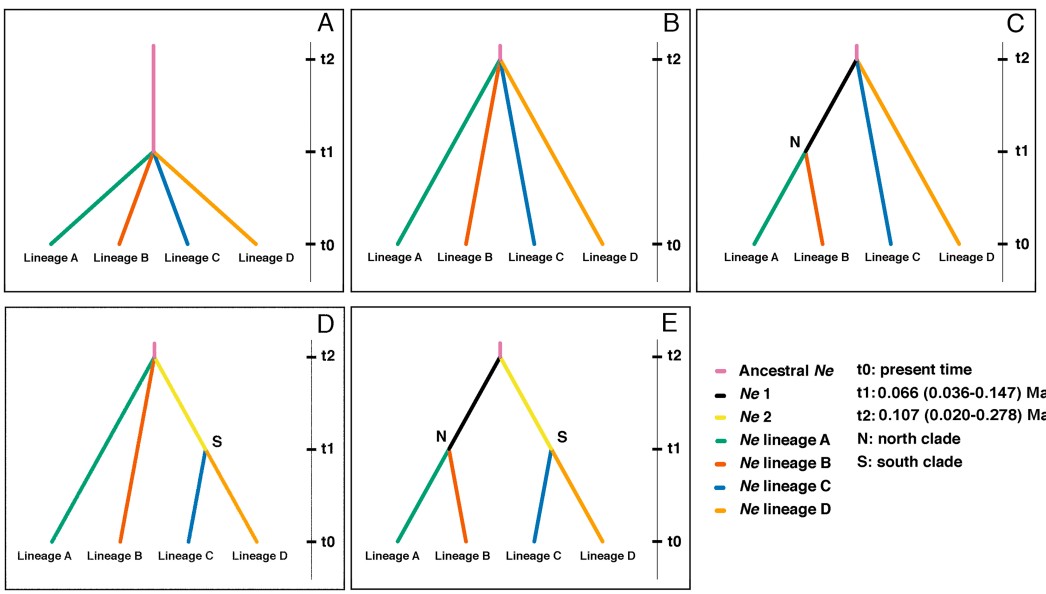

**Figure 5  Highest probable ancestral connectivity observed using Approximate Bayesian Computation analysis among the four lineages of *B. leptopus*.** (A) Scenario 1: as a null model all lineages coalesced at t1 in a single refuge. (B) Scenario 2: all lineage coalesced at t2 in a single refuge. (C) Scenario 3: starting at 0, lineage A coalesced at t1 in a single refuge with the lineage B. AB lineage ancestor coalesced at t2 with lineage C and D. (D) Scenario 4: lineage C and D coalesced at t1. CD lineage ancestor coalesced at t2 with lineage A and B in a single refuge. (E) Scenario 5: starting at 0, lineage A coalesced at t1 in a single refuge with the lineage B, same situation with lineages C and D. In the same way, AB and CD lineage ancestors coalesced at t2 in a single refuge.

**Table 4  Type I and Type II error rates and posterior probabilities for each scenario calculated from DIYABC.**

| True scenario used for simulation | Type 2 error rate | | | | | Type 1 error rate | Posterior probability (95% credible interval) |
|---|---|---|---|---|---|---|---|
| | Scenario 1 | Scenario 2 | Scenario 3 | Scenario 4 | Scenario 5 | | |
| Scenario 1 | – | 0.030 | 0.090 | 0.066 | 0.041 | 0.227 | 0.0438 (0.0000–0.4411) |
| Scenario 2 | 0.030 | – | 0.053 | 0.057 | 0.057 | 0.197 | 0.0044 (0.0000–0.4273) |
| Scenario 3 | 0.022 | 0.074 | – | 0.014 | 0.145 | 0.255 | 0.0000 (0.0000–0.4076) |
| Scenario 4 | 0.020 | 0.094 | 0.003 | – | 0.023 | 0.140 | 0.9339 (0.9057–0.9622) |
| Scenario 5 | 0.015 | 0.009 | 0.029 | 0.196 | – | 0.250 | 0.0178 (0.0000–0.4195) |

disjunct distribution for the species at the northern portion of the current distribution of *B. leptopus* (Fig. 6D).

## DISCUSSION

Our hindcasting-based approach supports the existence of four lineages in *B. leptopus* (A, B, C and D; Figs. 2 and 3) distributed discontinuously along the narrow, ~1000 km long Patagonian region of southern Chile. AMOVA confirmed the strongly different patterns of variation among the *Batrachyla leptopus* populations. In our study, most of the mtDNA

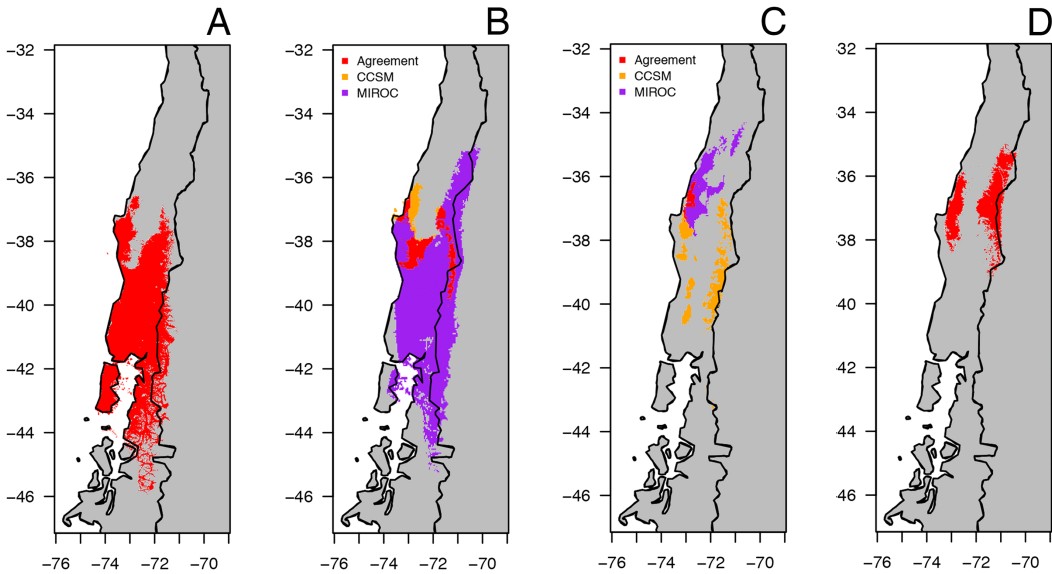

**Figure 6** **Geographical distribution for climatically predicted areas for the occurrence of *B. leptopus* based on current and past bioclimatic variables.** Potential distribution under: (A) current conditions (1960–1990). (B) Mid Holocene (Mid-Hol, ∼6,000 yrs BP). (C) Last Glacial Maximum (LGM ∼22,000 yrs BP). (D) Last Interglacial Period (LIG, ∼130,000 years before present). Red, orange, and purple represent areas with high probability of *B. leptopus* for each period.

genetic differences are present among localities and within lineages (Table 3), with less differentiation among lineages. This indicates the presence of high local genetic structure and high interpopulation differentiation. These results might suggest that although *B. leptopus* has a wide range of distribution, long-range dispersal is highly unlikely, which is in agreement with historical data. On the other hand, most of the variation in nuclear markers were observed within localities, while the values of both genes were not significant (Table 3).

The basic topology of the concatenated ML and Bayesian trees was similar, consequently we used the Bayesian tree as our primary hypothesis of relationships among *B. leptopus* populations (Fig. 3A). We recovered two main clades (named south and north) and four lineages (A, B, C, and D) strongly supported by boostrap and posterior probabilities. The nuclear phylogeny was unresolved (Figs. S1A, S1B) and provided no information on lineages relationships. In contrast, the mtDNA phylogeny (Fig. S1C) was highly informative, separating the same clades as the concatenated data set. Consequently, mtDNA variation was a leading indicator of population differentiation and phylogenetic relationships relative to nuclear loci in *B. leptopus*, indicating that mitochondrial markers provide more "signal" to track and higher mutation rates lineage divergence than any single nuclear gene, problably due to lower effective population sizes than nuclear genes (*Hung, Drovetski & Zink, 2016*).

## Biogeographic structure of the lineages

The two northernmost lineages (A and B) are currently separated by two major river systems (Itata and Bío Bío; Fig. 1); these boundaries also coincide with a region with sparsely distributed forest patches. These two lineages are genetically well-differentiated from each other (Fig. 2B); lineage A is restricted to the Los Queules Reserve, and lineage B is limited to a small area in Nahuelbuta range (Locality 2) near the Butamalal River. The evolution of distinct lineages or genetic clusters is often attributed to population isolation during glacial advances, which in combination with different selection pressures and/or genetic drift would drive population divergence (*Hewitt, 2004*). Similarly, low levels of current genetic diversity as observed in lineage B (Nahuelbuta; Table 2), suggest a recolonization history of founder effects, small population sizes, and genetic bottlenecks (*Hewitt, 2004*).

In contrast to these well-differentiated/low variability lineages, those in the southern distribution (lineages C and D) are geographically more heterogeneous. Lineage C is widely distributed from Máfil in the Los Ríos region (Locality 3 in Fig. 1) to Bahía Mansa, Los Lagos region (Locality 6 in Fig. 1), suggesting that certain landscape features, such as extensive forests, would have allowed dispersal among breeding groups over long timescales. Moreover, the encompassed areas of lineage C also include large mountains (e.g., Bahía Mansa and Cordillera Pelada; Fig. 1), suggesting that such orography could represent barriers to gene flow in *B. leptopus*, as reported in some co-distributed vertebrates and plants (*Sérsic et al., 2011*). The combined dataset suggests that lineage D is widespread throughout the rest of the species' range, with few phylogeographical subdivisions.

## Lineage divergence time

Divergence time estimates suggest that diversification of *B. leptopus* lineages may have occurred earlier than reported in other frogs such as the ground frog *Eupsophus calcaratus* (*Nuñez et al., 2011*), although co-distributed populations (e.g., Bahía Mansa) appear to have diverged later in time (0.025 mya for *B. leptopus* (Fig. 3A), and 0.065 mya for *E. calcaratus* (*Nuñez et al., 2011*; Fig. 2). *Vidal et al. (2016)* point out that some *B. leptopus* populations from Chiloé Island and the mainland (included in Lineage D in our study) diverged approximately 1.1 mya. Our results indicated that the initial split of *B. leptopus* into the North and South clades was during the Late Pleistocene (~0.107 mya; Fig. 3). Moreover, the overall pattern suggests that *B. leptopus* has undergone several rounds of fragmentation, followed by successive radiations within each clade. Further, at least two more recent series of fragmentation events are inferred within each of these clades. Our calibrations place the split between lineages A and B at ~0.068 mya, and between lineages C and D at ~0.065 mya.

The discrepancy in divergence times between our results and those of the *Vidal et al. (2016)* may be due to the use of different mutations rates (0.8%) and a single marker (mitochondrial *cyt b*). It is well known from population genetics theory that the stochastic nature of the genealogical process implies a significant amount of variance associated with parameter estimation. In fact, *Nabholz, Glémin & Galtier (2009)* suggest that divergence

inferences date should be based on statistical phylogenetic methods accounting for substitution rate variation across lineages.

Indeed, analysis of mtDNA sequence data can be enhanced in conjunction with nuclear sequences, which provide an independent estimate of phylogenetic relationships, mitigating the inherent stochasticity of genetic drift, and the variance associated with parameter estimation (*Carstens & Dewey, 2010*). Interpretations derived of the divergence time analyses also need to take into account the largely overlapping confidence intervals of the results for each lineage divergence. For example, if only the median values are considered, the results suggest coalescence of the lineages C and D at t1 (around 0.066 mya), and coalescence of the lineages A and B at t2 (around 0.107 mya). But if we consider the confidence intervals, the divergence of both clades, North and South could be closer in time than what is hypothesized in our reconstructions.

## Hypothesized refugia and post-glacial expansions

Studies on past contraction-expansion climate cycles in Patagonian landscapes suggest that rapid population expansions should occur in the biota affected by these processes, as habitats became more available (*Fraser et al., 2012*). Despite the geomorphological differences in Patagonian landscapes, population genetics theory predicts that a population undergoing rapid expansion should be characterized by low genetic diversity, since each new founder population represents only a fraction of the ancestral population (*Nichols & Hewitt, 1994*; *Hewitt, 2000*; *Hewitt, 2004*; *Waters, Fraser & Hewitt, 2013*).

The last two Pleistocene glaciations in southwestern South America (180 kya and 20 kya) covered the Andes with large ice fields reaching the Pacific Ocean south to 39°S, where the ice sheet decreased in elevation to sea level, and extending further to the southern tip of South America (*Rabassa, 2011*). Consequently, Late-Pleistocene divergence time estimates (0.107 mya) for first diversification of *B. leptopus* populations separating the North and South clades (Fig. 3), are consistent with a hypothesis of Pleistocene isolation followed by interglacial dispersal. In fact, the divergence of lineages A and B suggests that this interglacial dispersal from the ancestral population occurred rapidly across the current range of the species.

The existence of two suitable areas for the species is supported by the SDM for the LIG (0.120–0.140 mya) in that Los Queules population (Locality 1, Table 1) showed the highest genetic diversity (Table 2). This is typical for refugial populations that have been stable over time (*Fraser et al., 2012*). This evidence and the agreement of the two circulation models for Los Queules area as a suitable habitat for the species during the LGM (Fig. 6C), suggests that it is highly probable that the present Los Queules location is a remnant of the northern refuge, derived from the last southern Patagonian glaciation (180 kya).

Demographic reconstruction in *B. leptopus* using an ABC framework also supports the hypothesis of two putative refugia at different times during the Pleistocene (Scenario 4, Fig. 5D, Table 4). This scenario suggests that the South clade populations (lineages C and D) are likely descended from a divergence event approximately 65 Kya. This scenario is concordant with the predicted patchy distribution of the species during the Mid-Holocene (~6,000 years BP; Fig. 6B). On the other hand, the various demographic analyses show that

the genetic structure of lineages C and D contains signatures of demographic expansion consistent with Pleistocene glacial retreat. In fact, strong support for recent population expansion is represented by significantly negative Fu's Fs values, unimodal mismatch distributions with low raggedness indexes, and EBSP's depicting rapid expansion following the retreat of the Patagonian ice sheet after 15 Kya (Table 2; Fig. 4). This hypothesis is also reinforced with SDMs produced by both circulation models, although when viewed separately, the MIROC circulation model is the only one predicting this scenario.

## Past population dynamics and current conservation significance

During the Cenozoic climatic oscilations included multiple glaciations in southern South America (*Rabassa & Clapperton, 1990*; *Coronato, Martínez & Rabassa, 2004*). These geological events have been hypothesized as causes of the retreat and advance of temperate *Nothofagus* forests and conifers (*Villagrán & Hinojosa, 1997*; *Premoli, Kitzberger & Veblen, 2000*; *Tremetsberger et al., 2009*). Accordingly, several phylogeographic hypotheses suggest that Pleistocene glaciations had profound effects on the population genetic structure and variability of the Patagonian fauna. For example, glaciated populations of some fish species display molecular diversity (high haplotype diversity and low nucleotide diversity) significantly correlated with latitude (*Ruzzante et al., 2008*; *Cosacov et al., 2010*). The same genetic patterns have been observed in reptiles (*Breitman et al., 2011*; *Fontanella et al., 2012*), amphibians (*Nuñez et al., 2011*), and mammals (*Himes, Gallardo & Kenagy, 2008*; *Lessa, D'Elía & Pardiñas, 2010*).

In agreement with these previous studies, the location of glacial refugia and postglacial expansions identified here, indicate that the climatic niche of *B. leptopus* is likely to be related to an increase in the availability of suitable habitat in the southern part of its current distribution. Reconstruction of the potential distribution area of *B. leptopus* (Fig. 6) suggests that suitable habitats underwent expansions and contractions during glacial retreats and advances.

In addition to the inferences of past population dynamics, predictions about the ability of species to respond to future climate change play an important role in alerting potential risks to biodiversity. In fact, many studies have investigated the response of biodiversity to climate change, and most of them indicate that current and future rates of these changes may be too fast for ecological niches to evolve (*Fraser et al., 2012*; *Rolland et al., 2018*). This is particularly critical in species with low dispersal capacity such as amphibians, which makes them potentially less able to respond to changes induced by climate and, consequently, more vulnerable to extinction (*Duan et al., 2016*). In this context, the genetic impoverishment at northern area of the distributional range of *B. leptopus* is of great concern, given a climate change scenario based on increases in temperatures and aridity in central-southern Chile.

## CONCLUSIONS

Our study on genetic diversity throughout the geographic range of *B. leptopus* documents the existence of four lineages distributed along ~1000 km in southwestern Patagonia, including glaciated and non-glaciated areas during the LGM. The two northernmost

lineages are present in a region with poorly preserved forest patches, whilst the southern lineages are geographically more heterogeneous, suggesting that extensive forests would have allowed dispersion among breeding groups over multiple time scales. Late Pleistocene divergence estimates for the first diversification of the *B. leptopus* that separated the North and South clades, also supported by the SDM for the LIG, are consistent with a Pleistocene isolation followed by interglacial dispersion. The ABC analyses also supported the hypothesis of two putative refugia at different times during the Pleistocene, concordant with a patchy distribution of the species during the Middle Holocene. In addition, northern populations of *B. leptopus* showed the highest degree of isolation, these will require special attention given predicted increases in temperatures and aridity in south-central Chile; these populations are not inter-connected, and they could end up disappearing.

## ACKNOWLEDGEMENTS

The authors are grateful to Pablo Orozco Ter Wengel for his review and valuable comments on an early draft of the manuscript. We also thank three anonymous referees that significantly improved our work. We are grateful to Nicolás I. González for field assistance.

### Funding

This research was supported by NSF-PIRE OISE 0530267 to Jack W. Sites, Jr. and José J. Nuñez; Fondecyt 3160328 to Elkin Y. Suárez-Villota, and DID-UACH 2014-16 to José J. Nuñez. There was no additional external funding received for this study. The funders had no role in study design, data collection and analysis, decision to publish, or preparation of the manuscript.

### Grant Disclosures

The following grant information was disclosed by the authors:
NSF-PIRE OISE 0530267: 0530267.
Fondecyt: 3160328, DID-UACH 2014-16.

### Competing Interests

The authors declare there are no competing interests.

### Author Contributions

- José J. Nuñez and Elkin Y. Suárez-Villota conceived and designed the experiments, performed the experiments, analyzed the data, prepared figures and/or tables, authored or reviewed drafts of the paper, and approved the final draft.
- Camila A. Quercia performed the experiments, analyzed the data, prepared figures and/or tables, authored or reviewed drafts of the paper, and approved the final draft.
- Angel P. Olivares analyzed the data, prepared figures and/or tables, and approved the final draft.
- Jack W. Sites Jr conceived and designed the experiments, authored or reviewed drafts of the paper, and approved the final draft.

## Animal Ethics

The following information was supplied relating to ethical approvals (i.e., approving body and any reference numbers):

This study was carried out under supervision and approval of the Bioethics and Biosecurity Committee of the Universidad Austral de Chile (UACh, Resolutions No. 236/2015 and 61/15), and the Servicio Agrícola y Ganadero (SAG, Resolution No. 9244/2015). The Corporación Nacional Forestal, Ministerio de Agricultura, Gobierno de Chile allows to collect buccal swabs samples from wild protected areas (CONAF, Permit No. 11/2016.-CPP/ MDM/jcr/ 29.02.2016).

## Field Study Permissions

The following information was supplied relating to field study approvals (i.e., approving body and any reference numbers):

This study was carried out under supervision and approval of the Bioethics and Biosecurity Committee of the Universidad Austral de Chile (UACh, Resolutions No. 236/2015 and 61/15), and the Servicio Agrícola y Ganadero (SAG, Resolution No. 9244/2015).

## DNA Deposition

The following information was supplied regarding the deposition of DNA sequences:

The DNA sequences are available at GenBank: MK507982–MK508662.

## Data Availability

All sequences of the five markers obtained in this study are available as Supplemental Files.

## Supplemental Information

Supplemental information for this article can be found online at http://dx.doi.org/10.7717/peerj.9980#supplemental-information.

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
