# Peer review of "Phylogeographic analysis and species distribution modelling of the wood frog Batrachyla leptopus (Batrachylidae) reveal interglacial diversification in south western Patagonia"

_PeerJ, doi:10.7717/peerj.9980_

## Round 0.1 · original submission · Minor Revisions

Thank you for submitting your manuscript to PeerJ. I have sent your paper to three expert referees for their consideration. I have now received their comments back and have read through your paper carefully myself, and I really like it. Enclosed please find the reviews of your manuscript.

The reviews are in general favourable and suggest that, subject to minor revisions, your paper could be suitable for publication. Please consider these suggestions, and especially please pay attention to the questions raised by Reviewer 2, who did a very hard work on your manuscript and provided many useful comments. All raised questions must be fully addressed.

Personally, I would appreciate if you would add thumbnails showing life photos of your model species Batarchyla leptopus to Fig. 1 and Fig. 3 - this would make the figures look more attractive for a potential reader.

I look forward to receiving your revision soon.

Reviewer 1 ·

Basic reporting

In this study, Nuñez et al. explore the main phylogeographical patterns in the wood frog Batrachyla leptopus using a combination of the analysis of a multilocus dataset (3 mitochondrial genes + 2 nuclear loci) using both tradicional phytogeographical analyses as well as more modern ABC methods to test for specific hypotheses regarding patterns of population size and connectivity. These results were complemented by past and present species distribution modelling. The study is very well-written and the tested hypotheses are provided explicitly.

Experimental design

In general, both genetic and SDM methods agree with the best practices in their fields. The only exception perhaps is the mismatch distribution, given that the other used methods (EBSPs and ABC) have been shown to be vastly superior.

Validity of the findings

The conclusions are well-supported and provide an important contribution to the understanding of the evolution of species from southwestern Patagonia, as well as to the conservation of the wood frog.

Additional comments

A few minor comments:

L66: there is an extra comma before “have”
L92: patchy?
L202: Please provide information on the number of generations, their intervals, and parameter ESSs, as you indicated for the BEAST analysis on line 229-231.

Reviewer 2 ·

Basic reporting

This is an interesting and meaningful study focused at testing biogeographic hypotheses associated with Patagonian paleo-refuges, using as model the wood frog Batrachyla leptopus, with in-depth analyses covering phylogenetics, phylogeography, population genetics, divergence dating and ecological modelling. The tests are well-thought and the steps explained in a clear manner. Nevertheless, the conclusions are strongly based on an important bottleneck, the divergence dating analysis, which results are extrapolated for downstream ABC and niche modelling; because the authors use only a rate prior, extrapolated from a previous study, without checking for a a likely larger mitochondrial evolutionary rate, and without incorporating any fossils/biogeography as node date priors, their conclusions about the importance of past refuges should be taken with a grain of salt. Without further testing the effect of a faster evolutionary rate, no conclusions can be drawn temporaly, although spatial conclusions seem reasonable (population genetics, phylogenetic pattern).

Experimental design

Phylogenetics, phylogeography, population genetics, and ecological modelling analyses are all interesting and in line with what the authors aim to test.

However, the dating analysis may bear important biases. Because rate extrapolation is used without imposing fossil/biogeographic constraints, rate biasesshould have been considered, as the posterior times could be strongly affected by them (due to an inverse relationship between times and rates); and because times are an important issue in this ms, I strongly suggest an upwards correction (Nabholz et al., 2009) in BEAST, to check if correlation to paleo-refuges still holds. The Irisarri 2012 study apparently did not employ such a correction either.

Therefore, the divergence dating analysis assumes too much in its current configuration, and because downstream analyses assume the posterior time ranges reported as correct (when there's the possibility that they are not due to rate underestimation), it's very important that the authors re-run Beast with an alternative rate prior embracing (likely) larger rates for mitochondrial genes (as Nabholz et al. 2009 indicate that these are larger than commonly assumed).

Further questions/suggestions:

- The authors tested for saturation, but do not report if they found it for any marker in the results. In case there's saturation, older times could be underestimated, possibly inflating the tendency to infer recent population growth even if there is none (or if there is but it's an older one). If only a subset of the markers suggest a linear relationship with the best fit model, exclude the markers with more extreme signs of saturation, or else exclude positions with largest entropy (e.g., using Tiger or the likes), and re-run.

- Have the authors tested for clusterization by sampling tissue (i.e., swab vs. liver) in the phylogeny (or by genetic distances)? Given that there may be nuclear copies of mtDNA, or even multiple D-loop copies in a mtDNA (depending on the vertebrate group; and I couldn't find a complete Batrachyla nor Batrachylidae mtDNA genome at NCBI to testify for a single-copy D-loop in the species), it'd be important to show that there isn't such a clustered pattern of inferring an inexistent clade due to paralogy. Testing if clades are associated with tissue type shall give a hint at this.

- Regarding population structure, please also provide Fst measures by mtDNA concatenated, and by each nuclear locus (could be a SM table). Discuss if the 3 measures disagree and, if so, how such disagreement is in line or not with current conclusions.

- Do mtDNA concatenated vc nuc1 vs. nuc2 trees agree? Showing a SM figure with the 3 topologies would be enough. If they disagree, how does it impact conclusions?

- Does PhyML support partitioning? If not, how did the authors estimate the tree, assuming at least some partitioning had better fit than none (as often is the case)? If not, please run a partitioned ML using either RaxML, RaxML-ng, or IQTree.

- Were the 2 different nucDNAs concatenated in a single partition, or not? Please clarify. Also, tests of partitioning do not mind if nuc and mt partitions are mixed, but in fact they shouldn't be. If so, there's the chance that *BEAST run may be incorrect. There should be at least 2 partitions (mt plus 1-2 nuc partitions), and eventually more if the best partitioned model suggests so, e.g. by codon position within nuclear genes (regarded mt+nuc partitions are avoided). One nuc partition is acceptable if there's low nuclear variation.

- Still regarding the different types of loci in Beast: was the correct ploidy number assumed for mtDNA, and for nucDNAs?

Validity of the findings

Because the authors base their conclusions on the importance of paleo-refuges on dates that can be biased, it is hard to judge the validity of their findings in the current ms version.

Additional comments

A meaningful and important ms, using relevant techniques to test paleo-refuges using an anuran as model, but which needs further sensitivity analysis of divergence dating to become more credible. I recommend major revisions, mostly due to dating analyses.

Reviewer 3 ·

Basic reporting

The article Phylogeographic analysis and species distribution modeling of the wood frog Batrachyla leptopus (Batrachylidae) reveal interglacial diversification in south western Patagonia; It deals with a topic of particular importance to scientists interested in the evolutionary history of herpetofauna in southern South America. The particular case of the species of the Batrachyla genus, endemic to the Nothofagus forests, in southern Chile and Argentina has been of great interest by many scientists, who have revealed aspects of their natural history, characteristics, chromosomes, behaviors, larvae, etc. In this context, this article contributes significantly to a better understanding of aspects of the evolutionary history of one of its members, Batrachyla leptopus. This is a work very well documented in the extensive bibliography that exists on the matter. Furthermore, it is very well structured and therefore easy to read, even for someone who does not know molecular technologies in depth, but who, by understanding the context well, can realize that the objectives are well achieved and fully explained. For all of the above, this work deserves to be published without major corrections than those that could be derived from the very action of writing a scientific article.

Experimental design

The experimental design is appropriate for a job of this nature. The authors use three mitochondrial regions (D-loop, cyt b, and coI) and two nuclear loci (pomc and crybA1) to test their hypothesis. Indeed, taking into account that one of the questionable aspects of molecular methods is their robust statistics, this paper uses those most reliable methods, and incorporates a coalescence analysis, highly recommended nowadays as there is much confusion, due to the variability of results when a single molecular marker is used. On the other hand, the sample design incorporates specimens from the entire distribution, which makes the work very representative of the reality of B. leptopus. One aspect that could perhaps be lacking in this regard is some additional populations in the Andes Mountains, at 38 ° latitude (Curacautín) (Tohuaca National Park) and from north of Los Queules, but it does not diminish the importance of the work.

Validity of the findings

The results are valid since they are in line with other investigations carried out on the herpetofauna of southern South America, and have been obtained by applying an appropriate sampling design. Within the southern batrachofauna, the Batrachyla species, including Batrachyla leptopus, are part of a group of amphibians of tropical origin, and therefore their diversification may have occurred as a consequence of orographic processes subsequent to the current constitution of the continents. This is absolutely consistent with those expressed by the authors of this article. Furthermore, the historical antecedents that allow the evolution of South American amphibians to be contextualized do not provide many other additional hypotheses. On the other hand, the authors do not suggest in the text, if these four different lineages that appear in Batrachyla leptopus, present enough differentiation to be considered as possible different species. Perhaps it would be appropriate for the authors to discuss their results considering this hypothesis.
In this same line, It would be interesting to know, if the authors noticed any correspondence between lineages and some morphological distinction that suggests the presence of different taxonomic entities in Batrachyla leptopus. This question arises from the fact that during our extensive campaigns in the forests of southern Chile, we noticed some different morphs in B. leptopus, for example, some smaller specimens, the texture of their skin smoother.

Additional comments

I would like to congratulate the authors of this work for their important contribution to the knowledge of the evolutionary history of an important species in the herpetofauna of southern South America. One aspect that I would like to see at work is what consequences these results could have on the number of species in the genus. Could it be suggested that these four lineages represent four different species? If the molecular analyzes have a robust support, and the results obtained are consistent with other historical antecedents, can it be suggested that within species B. leptopus there is more than one species?
I would like the authors to discuss this point in some depth.
However, the authors mention that Batrachyla leptopus presents a very wide distribution (discontinuously ~ 1000 km) , it is important for the reader that the figures where maps are represented have a reference bar in km.
Do the authors have a photograph of a specimen from each of the lineages?

Annotated reviews are not available for download in order to protect the identity of reviewers who chose to remain anonymous.

---

## Round 0.2 · accepted · Accept

Thank you for taking the time to revise and resubmit your manuscript. I have now read through your paper as well as your letter in response to the reviews. I think that you have successfully addressed all of the concerns raised very well, and would like to accept your manuscript for publication in PeerJ. The reviewers also agree with this decision. Congratulations!

Reviewer 2 has provided some useful comments and corrections, I believe it would be possible to implement these at the stage of reviewing the proofs of your paper.

Thank you for all the hard work you have put into this. Your paper makes a strong contribution to the literature and I look forward to seeing it published.

Reviewer 1 ·

Basic reporting

My previous assessment of this study was already positive on the first round of reviews.

Experimental design

The experimental design follows the best practices in the field.

Validity of the findings

The conclusions are well supported by the data.

Additional comments

I believe that the authors addressed all of the issues I raised in my review.

Reviewer 2 ·

Basic reporting

The article used population, phylogeography, phylogenetics, divergence dating, and distributional modelling to study evolution of four populations of B. leptopus in the Patagonian region, arriving at relevant clues about association of their evolution to paleo-refuges.


*Conclusions section*

I think that the section "Conclusions", as it is now, is repetitive regarding what has already been said in Discussion. So a suggestion is to either remove it, or to use it expanding the discussion by comparing to the distribution of other organisms in the Patagonian region, and giving some perspectives of conservation and on what future studies could reveal further in terms of open questions.


*Text suggestions*:

Line 73: route that may have originated.

Line 86/87: re-structure phrase: "... that at least four Pleistocene glaciations occurred in the southwestern..."

Line 103: that tend to facilitate.

Line 122/123: Quaternary glaciations may have contributed to a phylogeographic.

Line 148: SDM approaches have.

Line 149: to assess species ranges, and to evaluate special.

Line 248: Due to the fact that. it is not possible.

Line 249: as there are no fossils available nor conspicuous dated biogeographic events.

Line 251: million years for the other mitochondrial.

Line 257: estimate divergence times.

Line 266: both approaches because small sample.

Line 296: corresponded to those.

Line 374: the latter with moderate support.

Line 376: Please add legend to Fig. S1, to which marker each tree corresponds to?

Line 441: probably due to lower effective.

Line 450: glacial advances, in which.

Line 452: could facilitate population divergence.

Experimental design

A few methodological doubts remain:

- one of the genes apparently was not included for dating, if so state it clearly, and why so.

- how comparison of models was done? Simply using marginal likelihoods is not acceptable, AICM is not as bad, stepping-stone is much better. If some other method, please clarify.

- does SH-test show that tree differences are related to low-support nodes? If not, best to say is that "visually" differed among non-supported nodes, or "probably" differed...

Validity of the findings

The steps taken are sensible, and the choice of tools as well. The results and discussion provide reasonable evidence for past paleo-refuges influencing population divergences within B. leptopus.

Reviewer 3 ·

Basic reporting

As I mentioned in my first review, the work of Núñez et al. Is an important contribution to the knowledge of the batrachofauna of southern South America. Batrachyla leptopus is a species with a very wide distribution throughout the temperate forest and also outside it. The samples analyzed in this work are representative of this wide distribution. Therefore, it is a very important contribution to the knowledge of this species, and could be understood in part, as an explanation of the morphological variability that is known about it. I think it is a great and valuable effort.

Experimental design

No comments on this section. I think authors explain totally this point to other referee.

Validity of the findings

I read the entire manuscript, and checked the tables and figures again. As well as the comments and responses to the other referees, and it seems to me that the methods used by the authors of the article are appropriate to the scientific question that they are trying to answer, therefore, in my opinion the results are valid and the findings absolutely relevant in context.

Additional comments

Again I want to congratulate the authors for their work, and I strongly urge them to try to go a little further in solving this type of problem (in another future article) about the specific problem. The boundaries between species are often confusing, and in my scientific opinion, it is necessary to establish clear positions regarding this fact. Mainly because this background is of the utmost importance when dealing with amphibian conservation issues.